# Five Years Monitoring the Emergence of Unregulated Toxins in Shellfish in France (EMERGTOX 2018–2022)

**DOI:** 10.3390/md21080435

**Published:** 2023-07-31

**Authors:** Zouher Amzil, Amélie Derrien, Aouregan Terre Terrillon, Véronique Savar, Thomas Bertin, Marion Peyrat, Audrey Duval, Korian Lhaute, Nathalie Arnich, Vincent Hort, Marina Nicolas

**Affiliations:** 1IFREMER (French Research Institute for Exploitation of the Sea)/PHYTOX/METALG, F-44311 Nantes, France; veronique.savar@ifremer.fr (V.S.); korian.lhaute@ifremer.fr (K.L.); 2IFREMER/LITTORAL/LER-BO, F-29900 Concarneau, France; amelie.derrien@ifremer.fr (A.D.); aouregan.terre.terrillon@ifremer.fr (A.T.T.); audrey.duval@ifremer.fr (A.D.); 3Laboratory for Food Safety, Pesticides and Marine Biotoxins Unit, French Agency for Food, Environmental and Occupational Health and Safety (ANSES), 94701 Maisons-Alfort, France; thomas.bertin@lisa.ipsl.fr (T.B.); marion.peyrat@anses.fr (M.P.); vincent.hort@anses.fr (V.H.); marina.nicolas@anses.fr (M.N.); 4Risk Assessment Department, French Agency for Food, Environmental and Occupational Health and Safety (ANSES), 94701 Maisons-Alfort, France; nathalie.arnich@anses.fr

**Keywords:** shellfish, brevetoxins, cyclic imines, cyanotoxins, BMAA, LC-MS/MS

## Abstract

Shellfish accumulate microalgal toxins, which can make them unsafe for human consumption. In France, in accordance with EU regulations, three groups of marine toxins are currently under official monitoring: lipophilic toxins, saxitoxins, and domoic acid. Other unregulated toxin groups are also present in European shellfish, including emerging lipophilic and hydrophilic marine toxins (e.g., pinnatoxins, brevetoxins) and the neurotoxin β-N-methylamino-L-alanine (BMAA). To acquire data on emerging toxins in France, the monitoring program EMERGTOX was set up along the French coasts in 2018. Three new broad-spectrum LC-MS/MS methods were developed to quantify regulated and unregulated lipophilic and hydrophilic toxins and the BMAA group in shellfish (bivalve mollusks and gastropods). A single-laboratory validation of each of these methods was performed. Additionally, these specific, reliable, and sensitive operating procedures allowed the detection of groups of EU unregulated toxins in shellfish samples from French coasts: spirolides (SPX-13-DesMeC, SPX-DesMeD), pinnatoxins (PnTX-G, PnTX-A), gymnodimines (GYM-A), brevetoxins (BTX-2, BTX-3), microcystins (dmMC-RR, MC-RR), anatoxin, cylindrospermopsin and BMAA/DAB. Here, we present essentially the results of the unregulated toxins obtained from the French EMERGTOX monitoring plan during the past five years (2018–2022). Based on our findings, we outline future needs for monitoring to protect consumers from emerging unregulated toxins.

## 1. Introduction

In France, according to European regulation, three groups of marine toxins are currently monitored: lipophilic toxins, saxitoxins (STX), and domoic acid (DA) [1]. In addition, numerous other groups of emerging toxins have been shown to be toxic, especially in mice by intraperitoneal injection, but are not yet regulated due to a lack of toxicological and epidemiological data to establish maximum levels in shellfish. In the context of this article, the authors define an “emerging toxin” as a toxin that is not regulated and for which there is little toxicological data available to characterize the hazard for humans; it can be a known toxin identified for the first time by a country’s monitoring network or a new derivative of a known toxin. In addition to risks linked to known marine toxic microalgae, other toxins suspected to be toxic to humans are being increasingly mentioned in the scientific literature as detected in shellfish: BMAA and freshwater cyanotoxins [2,3,4]. As filter feeders, shellfish accumulate toxins produced by harmful algal blooms (HABs), which can make them unsafe for human consumption. Algal species producing such toxins could be introduced into France via ballast water, biofouling on ship hulls, or commercial shellfish trade between countries. One of the main objectives of this work is, therefore, to carry out an inventory of the presence of these unregulated toxins in shellfish in France.

Cyclic imines constitute a group of lipophilic toxins sharing a macrocycle structure with a typical cyclic imine moiety, which appears to be the bioactive pharmacophore [5,6,7,8]. Indeed, the opening of these rings results in the loss of biological activity [5]. Currently, it is mainly spirolides (SPXs), gymnodimines (GYMs) and pinnatoxins (PnTXs), produced by *Alexandrium ostenfeldi*, *Karenia selliformis* and *Vulcanodnium rugosum*, respectively, that accumulate in shellfish [9,10,11,12,13,14]. All these compounds are neurotoxic to mice when injected intraperitoneally, but their toxicity to humans has not been proven. However, some are quite toxic orally to mice (e.g., pinnatoxins). Indeed, experimental data and knowledge about their mode of action lead to the conclusion that pinnatoxins may present a hazard to human health [12,15,16].

Palytoxin (PLTX), an amphiphilic polyether, is characterized on one side by the presence of numerous hydroxyls, amines, and amide groups, which make it hydrophilic, and on the other by a long lipophilic carbon chain. Palytoxin was first isolated in 1971 from *Palythoa toxica* (a tropical coral) [17,18] and is a Na^+^/K^+^ ATPase ion pump inhibitor [19]. Its analogs, called ovatoxins (OVTX-a to k, among which there is a strong dominance of OVTX-a), can also be produced by benthic dinoflagellates of the genus *Ostreopsis*. Among the described species, only a few are producers of toxic compounds, particularly *Ostreopsis* cf. *ovata* strains from different regions [20,21,22,23,24,25,26]. *Ostreopsis* blooms can affect human health through inhalation of aerosols [27] or direct skin contact [28]. *Ostreopsis* toxins can enter the food web and accumulate in several groups of organisms, including crustaceans, mollusks, fishes, and echinoderms [22,29,30,31,32].

Tetrodotoxin (TTX) is a low-molecular-weight polar neurotoxin. It binds to a receptor at site 1 on voltage-gated sodium channels, which blocks the channel [33]. Produced by marine bacteria (i.e., genus *Bacillus, Pseudomonas*, *Vibrio* [34])*,* the source of TTX in shellfish is still controversial and not definitively proven, although the dominant hypothesis is that it is of microbial origin [35]. More than 30 known analogs can be found in not only puffer fish but also marine gastropods and bivalves [36]. The parent toxin known to be lethal in humans has been identified in European shellfish in recent years [34,37,38]. The first TTX-related shellfish poisoning in Europe was reported in 2007, caused by a specimen of the gastropod *Charonia lampas* containing more than 300 mg TTX/kg in the digestive gland [33].

Brevetoxins (BTXs) are lipid-soluble cyclic polyether compounds known to be mainly produced by the dinoflagellate *Karenia brevis* [39]. The species *K. brevis* is recorded in Florida, the Gulf of Mexico, West Indies, and Oceania (New Zealand) [40]. Brevetoxin structures have a lactone function, which is considered necessary for their biological activity [41]. Human exposure mainly occurs through inhalation and ingestion. Brevetoxins can be transferred from water to air via aerosols, which may cause respiratory tract irritation in humans [42,43]. No human fatalities by inhalation related to BTXs have been reported. The presence of BTXs has been demonstrated in many species of shellfish, which are the main vectors of human food poisoning [13,16,44,45].

In addition to the contamination of seafood products by marine toxins, the literature also reports contamination of the marine environment by cyanotoxins [2,4,46], which could have a significant health impact. This phenomenon is likely to take place in estuarine environments, with cyanobacteria migrating from freshwater to marine environments. Microcystins (MCs), which are cyclic heptapeptides, are mainly hepatotoxic and toxic to the male reproductive system. Most studies on the presence of cyanobacteria and cyanotoxins have been conducted in freshwater ecosystems, but the intermittent physical transfer of MCs during freshwater discharge into coastal environments has recently been reported, for example, in California [47], Washington State [48], France [49], Lithuania [50], and Uruguay [51]. Anatoxins (ATXs), which are hydrophilic cyanotoxins, are secondary amine bicyclic alkaloids. A review on ATXs reported the occurrence of these cyanotoxins in water and biota, including fish, and in food supplements [52]. In addition, ATX-a in sea figs was associated with human food poisoning in France [53].

The neurotoxin β-N-methylamino-L-alanine (BMAA), a non-protein amino acid produced by terrestrial and aquatic cyanobacteria and by microalgae [54], has been suggested to play a role as an environmental factor in the neurodegenerative disease Amyotrophic Lateral Sclerosis-Parkinsonism-Dementia Complex (ALS-PDC), which occurred with high incidence in the South Pacific island of Guam in the 1950s [55,56]. The ubiquitous presence of BMAA in aquatic environments and organisms along the food chain potentially makes it a public health concern [3,57,58]. In France, BMAA was detected in shellfish for the first time in Thau Lagoon in the south of France [59]. These authors also formulated the hypothesis that this contamination could be related to an ALS cluster in that area. These results led the French Ministry to refer the matter to the French Agency for Food, Environmental, and Occupational Health and Safety (ANSES), which identified BMAA as a hazard to human health [60].

The EMERGTOX network has been in place since 2018 to systematically monitor these unregulated toxins throughout the year and highlight emerging toxins in shellfish. It was intended to complement the existing French regulatory monitoring program for marine toxins in shellfish during toxic microalgal blooms (REPHYTOX), which is dedicated to regulated toxins in Europe. The main objective of the EMERGTOX network is to acquire data on the main groups of unregulated lipophilic and hydrophilic toxins listed at the international level in order to contribute to risk assessment. The analytical approach used also allows the monitoring of regulated toxins and consequently the possible identification of hazards related to the presence of these toxins in shellfish outside favorable periods of toxic algal blooms (e.g., winter), as a complement to the official monitoring. Indeed, three analytical approaches using mass spectrometry were optimized and in-house-validated to monitor all targeted lipophilic toxins: (i) okadaic acid/dinophysistoxins, azaspiracids, yessotoxins, domoic acid, in addition to lipophilic toxins not regulated in Europe (pectenotoxins, spirolides, gymnodimines, pinnatoxins, ovatoxins/palytoxins, brevetoxins, microcystins, and nodularin); (ii) hydrophilic toxins: saxitoxin group and unregulated toxins (tetrodotoxins, anatoxins, cylindrospermopsins); and (iii) β-N-methylamino-L-alanine (BMAA), 2,4-diaminubutyrique acid (DAB) and N-(2-aminoéthyl)glycine (AEG), which are also not regulated. This article aims to summarize the results obtained on unregulated lipophilic and hydrophilic toxins of marine and freshwater origin in shellfish production areas along the French coasts during the period 2018–2022.

## 2. Results

### 2.1. LC-MS/MS Lipophilic Multi-Toxin Quantification in Shellfish

#### 2.1.1. Optimization and Internal Validation of the LC-MS/MS Lipophilic Toxin Analysis Method in Shellfish

The toxins targeted here are described in the Section 4. A new analytical approach was developed to quantify 44 toxins (regulated and unregulated) in shellfish samples. This approach required the use of three LC-MS/MS methods (i.e., three elution gradients and two ionization modes). Chromatograms obtained for each of the three analytical methods are given in Figure 1a–c, showing the performance of the analytical approach.

Single-laboratory validation of this approach was carried out according to the NF V03-110 [61] standard through the construction of an accuracy profile. Excellent performance was obtained with acceptance limits in the range of 15–25% [62]. Such performances are very satisfactory. The LC-MS/MS results of the field samples analyzed are corrected by ionization and extraction recovery (Appendix A). This broad-spectrum method allowed quantifying several regulated (OA/DTXs, YTXs, AZAs, DA) and non-regulated (PTXs, SPXs, PnTXs, GYMs, PLTX/OVTXs, MCs, NOD) lipophilic toxin groups. The LOD and LOQ of each of the toxins are given in Appendix A. Here, we present results on unregulated lipophilic toxins.

#### 2.1.2. Targeted LC-MS/MS Screening of Lipophilic Toxin Groups Monitored by the EMERGTOX Network

Firstly, neither the palytoxin/ovatoxin (PLTX/OVTX) nor the nodularin (NOD) groups have been detected in filter-feeding bivalve mollusk samples since the EMERGTOX network was set up in 2018. All the results obtained show that, in addition to the detection of groups of regulated toxins (OA/DTXs, YTXs, AZAs, and DA) (results not shown), groups of lipophilic marine toxins unregulated in Europe and freshwater microcystins were detected at low concentrations in marine shellfish in France (SPXs, GYMs, PnTXs, BTXs, MCs).

Figure 2 shows the percentage of shellfish samples containing the unregulated lipophilic toxin groups during the period 2018–2022. The annual spatial and temporal evolution of the percentage of each of the targeted toxin groups remained generally similar during this period, except for SPXs and MCs, for which there was a decrease of 20% between 2018 and 2022.

The highest percentages were for spirolides (SPX-13-DesMeC, SPX-DesMeD), which were systematically present in shellfish at all EMERGTOX sites. Toxins of the pinnatoxin group (PnTX-G or/and PnTX-A) were present in 20–28% of samples, mainly in the Mediterranean area. The microcystin group (MC-RR or/and dmMC-RR or/and MC-LR), present in shellfish at five sites on the English Channel and the Atlantic coast, was found in 5–15% of the samples. The very low percentages correspond to toxin groups that were only found at a few sites: gymnodimine (GYM-A) at two sites (one on the Atlantic coast, one on the Mediterranean coast (Corsica); and brevetoxins (BTX-2, BTX-3), only found on the Mediterranean coast (Corsica). Figure 3 shows the presence of the maximum concentrations of unregulated lipophilic toxins at different sites during 2018, the first year of operation of the EMERGTOX network: SPXs, PnTXs, GYMs, MCs, and BTXs. This was the first time the BTX group (BTX-2, BTX-3) was detected in France and Europe.

The same qualitative profile for unregulated toxins was found in shellfish at the EMERGTOX sites in the following years, between 2019 and 2022 (Appendix A). These results show a recurrence of the same groups of unregulated lipophilic toxins on the Atlantic and Mediterranean coasts of France.

For the monitoring of pinnatoxins in the Thau Lagoon in the Mediterranean Sea, following depletion of the mussel stock at the Ingril site in early 2021, we continued monitoring at the Vic site located in the same area of the Thau Lagoon using another shellfish taxon (clams instead of mussels).

All the results obtained on shellfish samples collected at different sites of the EMERGTOX network show five types of unregulated toxin profiles in shellfish along the French coastline (Figure 2 and Figure 3). Firstly, all the sampled sites showed the systematic presence of spirolides at low concentrations during the period 2018–2022, although higher levels were measured on the Atlantic coast and on the Mediterranean coast, with a maximum value observed in mussels from Le Scoré in July 2021 (19 µg/kg TF). In addition to SPXs, (i) MCs were detected every year at two sites on the English Channel and on the Atlantic coast. The maximum level was quantified in February 2022 in mussels (9 µg/kg TF) collected at Kervoyal on the Atlantic coast; (ii) GYMs were detected at two sites on the Atlantic coast (Le Scoré, Arguin) and one on the Mediterranean coast (Corsica), with a maximum in mussels from Corsica in July 2021 (3,5 µg/kg TF); (iii) PnTXs were detected on the Atlantic coast (Le Scoré) and at four sites on the Mediterranean coast (Marseillan, Ingril, Vic, Corsica), with a maximum in mussels from the Ingril site in October 2018 (473 µg/kg TF); (iv) BTXs were detected on the Mediterranean coast (Corsica), with a maximum in mussels in November 2020 (57 µg/kg TF).

Furthermore, to have a wider view of the contamination and to diversify the shellfish taxa sampled, a gastropod (whelk) was added to the EMERGTOX network in 2021. No lipophilic toxin belonging to the targeted toxin groups was detected in any samples in 2021 or 2022.

### 2.2. HILIC-MS/MS Quantification of Saxitoxin, Tetrodotoxin, Anatoxin, and Cylindrospermopsin Groups in Shellfish

#### 2.2.1. Optimization and Internal Validation of the HILIC-MS/MS Analysis Method

The hydrophilic toxins targeted here are described in the Section 4 section. The method has been fully in-house validated using matrix-matched calibration curves. The performance characteristics studied were deemed satisfactory. For most of the toxins, when standards were commercially available, mean recovery rates ranged between 80% and 108%. Only doCYN recovery was lower, with a value of 72%. Appendix A shows the overall recovery of each toxin. Repeatability was at worst 15%, and intermediate precision was less than 20%. Figure 4 gives the chromatograms obtained for each of the two gradient conditions of the analytical methods, showing the performance of the analytical approach. The LOD and LOQ of each of the hydrophilic toxins are given in Appendix A.

#### 2.2.2. HILIC-MS/MS Analysis of Saxitoxins, Tetrodotoxins, Anatoxins, and Cylindrospermopsin in Shellfish

The monitoring of these hydrophilic toxins in the EMERGTOX network started in January 2019. Hydrophilic-regulated toxins (STXs) and unregulated toxins (TTXs, CYNs, and ATXs) were sought through the HILIC-MS/MS approach. The results obtained show, in addition to the detection of STXs (results not shown), the presence of ATX and CYN in mussels for the first time in France.

Cylindrospermopsin was identified at Ingril from March to July 2019, in December 2019, and in January 2020; at Marseillan in July 2019 and June 2021; and at Le Scoré in October 2020 and July 2021. Anatoxin was detected for the first time in mussels in February 2019 at the Ingril site and in March 2021 at the Le Scoré site.

One of the major findings in the monitoring of unregulated hydrophilic toxins was the recurrent presence of CYN, detected for the first time on the Mediterranean coast (Thau Lagoon) between March 2019 and January 2020, with a maximum level of 18 µg CYN/kg TF during the summer of 2019 (Figure 5).

Furthermore, it is worth highlighting an atypical result concerning the regulated hydrophilic toxins monitored since 2019. This is about the recurrent presence in winter of toxins of the saxitoxin group (GTX-2, -3; GTX-1, -4; STX) in Arguin oysters on the Atlantic coast, with a maximum of 100 µg GTX-2/kg TF in January 2022 (Figure 6). No known toxic microalga species that produce these toxins were observed in situ during this period, which was not favorable to algal blooms.

### 2.3. HILIC-MS/MS Quantification of BMAA and Its Analogues in Shellfish

#### Optimization and Internal Validation of the HILIC-MS/MS Analysis

For the screening of BMAA and its analogs (DAB, AEG), we used the analytical method optimized in-house [54]. Using a factorial design, we optimized and characterized the chromatographic separation of BMAA and its analogs by hydrophilic interaction chromatography coupled to tandem mass spectrometry (HILIC-MS/MS). A combination of an effective solid phase extraction (SPE) clean-up, appropriate chromatographic resolution, and the use of specific mass spectral transitions allowed us to develop a highly selective and sensitive analytical procedure to identify and quantify BMAA and its analogs in shellfish matrices. Appendix A gives the average recovery of BMAA and its analogs. The use of D_3_BMAA and D_5_DAB as the internal standards allowed correction and accurate quantification of BMAA and its analogs in all the samples. For BMAA and DAB, the limit of detection (LOD) equaled the limit of quantification (LOQ) and was 38 µg/kg of whole shellfish flesh. Figure 7 shows chromatograms illustrating the performance of this analytical approach.

While AEG was not found in any of the samples, a systematic year-round presence of BMAA and DAB was found in all shellfish samples analyzed as part of the EMERGTOX network (mussels, oysters, clams, and whelk). Figure 8 shows an example of the evolution of the monthly concentration of BMAA and DAB in mussels from the Kervoyal site (Atlantic coast) during 2019–2022 (samples could not be collected in January, February, or April 2020 due to the COVID epidemic). The “annual distribution” of the toxin profile is similar from year to year.

Figure 9 shows the maximum concentrations of BMAA and DAB in shellfish per year and per site monitored over the period 2019–2022 (except for the Vic site, which started in 2020; the “Ouest Baie de Seine” and “Baie des Veys” sites started in 2021). The annual evolution of BMAA and DAB concentrations was relatively similar from one year to the next for each of the sites during the 2019–2022 period.

For all the sites, the maximum BMAA content found was about 10,000 µg/kg whole flesh (WF) in the mussels at Kervoyal on the Atlantic coast (about 70% of the total BMAA + DAB) in 2021. Concerning the accumulation in oysters, the maximum BMAA content of about 7000 µg/kg WF (about 80% of the total) was observed at the Marseillan site on the Mediterranean coast in 2021. For clams collected at Vic on the Mediterranean coast (Thau lagoon), the maximum concentration was about 5000 µg/kg (about 70% of the total) in 2021. For whelks sampled on the English Channel, the maximum reached was about 3000 µg/kg of WF in 2021 (about 70% of the total).

## 3. Discussion

### 3.1. Contamination of Shellfish by Unregulated Lipophilic Toxins

The results obtained by the EMERGTOX network since it was set up in January 2018 show that unregulated lipophilic toxins (SPXs, PTXs, PnTXs, GYMs, BTXs, MCs) have been quantified every year in shellfish in France (various species of bivalve mollusks and whelks as representative of the gastropods). Some of these toxins were found for the first time in some bivalve mollusk production areas during this period: (i) microcystins (MC-RR and/or dmMC-RR and/or MC-LR) mainly during autumn/winter; (ii) gymnodimine (GYM-A) during summer; and (iii) BTXs (BTX-2, -3) during autumn/winter. In contrast to filter-feeding bivalve mollusks, no lipophilic toxins belonging to the toxin groups targeted in this work were detected in whelk (gastropod) samples.

In Europe, SPX was found in shellfish in Norway [9], Spain [63,64], and Italy [65], whereas PnTX was found in mussels in various parts of the Norwegian coast [13,66]. Recently, a three-year survey (2019–2021) was carried out in Greece on the contamination of shellfish by cyclic imines, implementing an LC-MS/MS method. Out of 6911 samples analyzed, gymnodimines, spirolides, and pinnatoxins were quantified in 1123, 585, and 3085 samples, respectively. The maximum concentrations found were 74, 69, and 64 µg/kg for GYM, SPX, and PnTXx, respectively (Oral communication by Dr Maria Kalaïtzidou, Head of NRL for Marine Biotoxins, Thessaloniki).

España Amórtegui et al. [67] investigated LC-MS/MS 17 cyanotoxins (13 MCs, NOD, ATX-a, h-ATX, CYN) in 13 mussel and oyster samples collected during the summers of 2020–2022 along the Bohuslän coast (Sweden). Nodularin was quantified in all shellfish samples analyzed in the range of 7–397 μg/kg.

Among the unregulated lipophilic toxins regularly detected by the EMERGTOX network, some had already been identified for the first time as part of the vigilance program set up for the chemical analysis of lipophilic marine toxins alongside the mouse bioassay, which was the reference test before 2010: spirolides (SPX-13-DesMeC, SPX-DesMeD) since 2006 [10], pinnatoxins (PnTX-G, PnTX-A) since 2010 [14]. In addition to these toxins, EMERGTOX has been able to detect brevetoxins (BTX-2, BTX3) since 2018 [45], microcystins (MC-RR, dmMC-RR, MC-LR) since 2018, and gymnodimine (GYM-A) since 2021. The demonstration of microcystins and gymnodimine in French marine shellfish has not yet been published.

Although toxins of the palytoxin group (PLTX/OVTX) were not detected in shellfish monitored by the lipophilic multi-toxin analytical approach used in the EMERGTOX network, this group of toxins has been detected (with lower LOD/LOQ values) in other seafood organisms (e.g., sea urchins, fish, gastropods, crustaceans) in French Mediterranean coastal areas (without bivalve mollusk production sites that can be monitored as part of EMERGTOX). This recurrent contamination since the 2000s, mainly by ovatoxins (OVTXs), has been associated with summer blooms of *Ostreopsis* cf. *ovata* [22,26,30,31,68].

In addition to the presence of lipophilic phycotoxins produced by marine microalgae, microcystins (MC-RR, dmMC-RR) produced by freshwater cyanobacteria were also detected in samples of marine bivalve mollusks (mussels, oysters), particularly during winter on the English Channel and Atlantic coasts (Baie de Vilaine, Charente Maritime). The environmental origin of microcystins in marine bivalves in winter in France is not known. In the absence of a *Microcystis* bloom, bivalves can also accumulate MCs from sediments. Indeed, a recent study showed that the sediments of a delta river contained MCs even during periods not favorable for a typical bloom [69]. The MCs could originate from planktonic cyanobacteria sinking in the sediment or from benthic MC-producing cyanobacteria. In a related study, we showed the presence of intact *Microcystis* colonies with *mcyB* genes and intracellular MCs in Atlantic Ocean sediments [70]. Cyanotoxins are usually regularly monitored in fresh waters but rarely in estuarine or marine waters, despite the possibility of their export downstream. Indeed, as part of an experimental research project on the fate of cyanobacteria along the land/sea continuum, we demonstrated that there was a physical transfer of dissolved microcystins over about 10 km along a river from a freshwater reservoir affected by summer blooms of toxic cyanobacteria to the south Brittany coast. Indeed, MCs were quantified in caged mussels and oysters placed at several sites along the freshwater/estuary/seawater continuum [4,49]. The extracellular MC content increases with the salinity gradient following the lysis of cyanobacterial cells due to osmotic shock. These dissolved MCs are then found in estuarine and marine mollusks. In a similar way to the EMERGTOX results, our field study during the summer bloom of *Microcystis* showed that marine mussels accumulated more dmMC-RR than MC-RR [4]. We found that the proportion of dmMC-RR increased significantly in mussels (3–50%) with increasing salinity. They hypothesized that MC-RR, which are abundant in *Microcystis*, are metabolized into dmMC-RRs in the mussel tissues. Although MCs could have a significant impact on human health, the level of contamination in seafood consumed by humans may have been underestimated, as only MCs in their free (non-protein-bound) form are commonly quantified in tissues. Indeed, after ingestion by marine organisms, MCs remain free and/or covalently bind to protein phosphatases (PP2A) [71]. It has been shown that the largest fraction of MCs bound to proteins is found in tissues and that these are eliminated slowly [72,73]. However, the bioavailability (and toxicity) of bound MCs in mussel tissues ingested by humans is still not known.

As for BTXs, in the frame of the EMERGTOX network, only BTX-2 and BTX-3 are analyzed by lipophilic multi-toxin LC-MS/MS. This suggests that the concentrations of BTXs found are underestimated since the representative toxins of the BTX group have not all been investigated [44,74]. Consequently, in addition to the targeted LC-MS/MS method that uses the few commercially available BTX standards, a complementary approach is needed to take into account the BTXs for which standards are not yet commercially available but which are relevant with regard to food safety. For example, the use of an ELISA test [75] allows for the screening of both BTXs in microalgae and metabolites formed in shellfish flesh. The first results obtained by ELISA on BTX-contaminated mussel samples harvested in Corsica in 2020 revealed a higher concentration of equivalent BTX-3 than that found by LC-MS/MS. This implies the presence of brevetoxin analogs other than the BTX-2 and BTX-3 quantified by LC-MS/MS in the framework of EMERGTOX. Based on the toxicity data, a priority list of analogs to be searched for has been established by ANSES and includes the following metabolites: BTX-1, BTX-2, BTX-3, BTX-B3, BTX-B1, BTX-B2, S-deoxy-BTX-B2, N- myristoyl-BTX-B2, and N-palmitoyl-BTX-B2 [76]. We are, therefore, developing a targeted chemical analysis method specific to brevetoxins using LC-MS/MS as well as a complementary approach using ELISA to monitor brevetoxin metabolites, including those for which standards are not available. A non-targeted analysis will also be implemented using high-resolution mass spectrometry (LC-HRMS) to provide spectral information to identify possible brevetoxin metabolites detected by ELISA but undetected by the targeted LC-MS/MS analysis.

### 3.2. Contamination of Shellfish by Unregulated Hydrophilic Toxins (ATXs, CYNs)

Tetrodotoxins (TTXs) were not detected in the EMERGTOX framework probably because of the high LOD of the multi-toxin method used without a purification step on the one hand and the low concentrations observed so far in France [37,38]. In Europe, TTX was found in shellfish harvested in Dutch production areas in 2015, 2016, and 2017. Of 1063 samples analyzed, the highest concentrations were observed in 2016: 253 μg TTX/kg in oysters and 101 μg TTX/kg in mussels. The presence of TTX seems constant over the last three years, with the highest concentrations observed each year at the end of June [77]. Very few data are available regarding the accumulation of unregulated hydrophilic cyanotoxins (CYNs and ATXs) in marine shellfish. The Australian freshwater mussel (*Alathyria pertexta*) can accumulate between 130 and 560 µg CYN/kg FW [78]. The accumulation of CYN in the freshwater mussel *Anodonta cygnea* was observed after 16 days of exposure to cylindrospermopsin-producing cultures of *Cylindrospermopsis raciborskii* [79]. ATX-a was identified in sea figs associated with human food poisoning in France [53]. The origin of these cyanotoxins found in French shellfish on the Atlantic (Le Scoré) and Mediterranean coasts (Marseillan, Ingril) remains unknown.

Further to the monitoring of unregulated toxins, EMERGTOX can detect the presence of regulated toxins in shellfish outside periods at risk of toxic algae blooms, particularly during the winter. In contrast, the REPHYTOX regulatory monitoring network, implemented during toxic algal bloom periods, does not allow such observations. This is the case for the systematic detection of the saxitoxin group (GTX-2, -3; GTX-1, -4; STX) in oyster samples from the Arguin site in winter since the establishment of the EMERGTOX network in 2019, whereas no observation of known toxin-producing microalgae was reported in the phytoplankton monitoring network. In this case, the environmental origin of the STX group is still unknown.

### 3.3. Contamination of Shellfish by BMAA

The EMERGTOX results obtained reveal the continuous presence of BMAA and DAB in all the shellfish samples analyzed (mussels, oysters, clams, whelks) from all the French coasts monitored, for which the potential risk for consumers is currently unknown. Concentrations of BMAA in shellfish can vary according to the species, the trophic status of BMAA producers, and the different levels of environmental contamination depending on geographical and seasonal factors. The relatively constant concentration of DAB in shellfish suggests that the DAB molecule could be present in primary producers or bacteria and/or be naturally present in shellfish flesh. Their environmental origin and metabolism in mollusks are still unknown.

As part of Ifremer’s previous work, we demonstrated the presence of BMAA and its analog DAB in all mollusks collected (e.g., mussels, oysters, scallops) during 2013 in shellfish growing areas along the French coasts (English Channel, Atlantic Ocean, Mediterranean Sea) as well as in mussels from other countries along the Mediterranean Sea (e.g., Italy, Spain). Only small variations in BMAA and DAB levels were observed, and these were not correlated with any of the phytoplankton species reported [80,81].

It appears that the presence of BMAA is widespread in mollusks. Indeed, almost all studies indicate that mollusks contain BMAA (and its isomers, when included in the analysis). The levels of BMAA observed in France are similar to those found in other studies conducted elsewhere [3]. Salomonsson et al. [82] analyzed shellfish marketed in Stockholm and reported BMAA concentrations of 0.08–0.9 mg kg^−1^ FW in mussels and oysters from the Swedish west coast and 0.32 mg kg^−1^ FW in oysters from Greece. In Portugal, concentrations of up to 0.08 mg kg^−1^ FW of BMAA have been reported in cockles. The two isomers AEG and DAB were also detected in cockle tissue but were not quantified [83]. Additionally, epidemiological studies are needed to assess the potential link between high local consumption of contaminated mollusks and some large clusters of Amyotrophic Lateral Sclerosis (ALS) or other chronic neurological disorders. One route of human exposure may be the consumption of aquatic organisms containing BMAA, as has been demonstrated for shellfish and fish in fresh, brackish, and marine waters worldwide [59,80,81,84,85]. Based on the recent assessment by ANSES, the hypothesis of BMAA exposure as a factor promoting neurotoxicity and neurodegeneration is considered highly likely [60].

### 3.4. Guidance Level for Unregulated Toxins in French Shellfish

In the absence of maximum levels for toxins not regulated in Europe, guidance levels were set by the European Food Safety Authority (EFSA) for the palytoxin/ostreocin (PLTX+OST-D) and tetrodotoxin (TTX) groups: (i) 30 µg (PLTX+OST-D)/kg shellfish meat [86]; (ii) 44 µg TTX/kg shellfish meat [34], respectively. At the French level, ANSES set guidance levels for some unregulated toxins identified for the first time in France, based on a protective default portion size of 400 g shellfish meat (similar to EFSA methodology): 23 µg PnTX-G equivalent/kg shellfish meat for the pinnatoxin group [16], 180 µg BTX-3 equivalent/kg shellfish meat for the brevetoxin group [87], and 15 µg PLTX equivalent/kg shellfish meat for the sum PLTX + OVTX + OST-D [88]. Note that BTXs are not regulated in Europe but are in the United States, Mexico, and New Zealand (maximum permissible level of 800 µg BTX-2 equivalent per kg of shellfish meat) [89,90].

## 4. Materials and Methods

### 4.1. Materials

Materials included methanol (LC/MS grade), acetonitrile, MilliQ water, ammonium formate (grade: for LC-MS), concentrated formic acid 99–100% (LC/MS grade), sodium hydroxide[2,5] (grade: for analysis), hydrochloric acid [5 M] (grade: for analysis), 1% acetic acid, NH_3_ (LC-MS grade), trichloroacetic acid (TCA), 0.1 M ammonia and cartridges of amorphous polymer graphitized carbon.

Commercially available toxin standards (National Research Council, Halifax, Canada; Novakits): pectenotoxins (PTX-2); spirolides (SPX-13-desMe-C); gymnodimines (GYM-A); pinnatoxins (PnTX-G, PnTX-A); brevetoxins (BTX-2, BTX-3); palytoxin (PLTX); microcystins (dmMC-RR, MC-RR, MC-LA, MC-LF, MC-LY, MC-LW, dmMC-LR, MC-LR, MC-YR); nodularin (NOD-R); anatoxins (ATX); cylindrospermopsins (CYN, doCYN); saxitoxin group; tetrodotoxin (TTX), D_3_BMAA, and D_5_DAB.

Standards for the following toxins are not available: PTXs (PTX-1, PTX-2-sa, PTX-2-sa-épi, PTX-6); SPXs (SPX-desMe-D, SPX-13,19-didesMe-C); GYM-B; PnTX-E, PnTX-F; OVTX-a; hATX; TTXs (épi-TTX, 4,9 anhydro-TTX, 5,6,11-trideoxy-TTX, 5-déoxy-TTX, 11-déoxy-TTX). Therefore, these toxins have not been validated intra-laboratory. The results of these toxins in the samples analyzed are given as the equivalent of the available toxin standard belonging to the same toxin group.

### 4.2. Sampling Shellfish

The quantification of target toxins was carried out on different shellfish taxa: mussels (*Mytilus edulis*, *Mytilus galloprovincialis*), oysters (*Crassostrea gigas*), clams (*Ruditapes decussatus*), and whelks (*Buccinum undatum*). These shellfish, weighing a total of 2 kg, were obtained monthly and collected at 12 sites along the French coasts of the English Channel, the Atlantic Ocean, and the Mediterranean Sea (Figure 10; note: between 2018 and 2020, only 11 sites were used). These shellfish sampling sites were selected in collaboration with the risk assessment department at ANSES. The criteria for choosing the sites were as follows:Their location in shellfish harvesting areas (mussels, oysters, clams, or whelks) that are active all year round;An equal geographical distribution over the French coastlines;The existence of historical data for the sampling points, with the aim to obtain long time series;Previous mouse bioassay results for lipophilic toxins (which was the reference test before 2010) that remain unexplained (short survival time and/or neurological symptoms);Their location outside areas affected but not immune to possible contamination by potentially toxic microalgae.

Analysis of lipophilic toxins in the digestive glands gives better detection results when the target toxins are present at low concentrations because the digestive glands concentrate more on trace compounds. For mussels and oysters, the concentrations found were converted to the amount in the total flesh (TF) by calculation, using the actual percentage (%) of DG to total flesh (M_DG_/M_TF_ × 100): T_TF_ = T_DG_ × % DG. Extrapolation of the results obtained on the DG to the total flesh assumes that almost all lipophilic toxins are concentrated in the DG.

The analysis of hydrophilic toxins (STX, TTX, BMAA, ATX, CYN) was performed on the total flesh of shellfish.

Each raw shellfish sample (total flesh “TF” or digestive gland “DG”) was drained, ground, and homogenized.

### 4.3. Methods

#### 4.3.1. LC-MS/MS Analysis of Lipophilic Toxins and Domoic Acid

##### Sample Preparation

The different steps of the preparation of shellfish extracts were optimized by using blanch matrices spiked with the targeted toxin standards. From the homogenate of 2 kg of harvested shellfish, a subsample of 200 ± 5 mg was placed in a 2-mL Eppendorf tube containing 250 ± 5 mg of glass beads with a diameter of 100–250 µm for mussels, oysters, and clams and 750–1000 µm for whelks; subsequently, 945 µL of methanol (MeOH) was added. The sample was ground using Mixer Ball Milling equipment (MM400, Retsch) for 2 min for mussels, oysters, and clams and 5 min for whelks at 30 Hz, then centrifuged for 5 min at 15,000× *g* at 4 °C. The supernatant was then transferred to a 2 mL volumetric flask. This operation was repeated, then the two supernatants were combined in the 2 mL flask and the volume was adjusted to 2 mL with MeOH. The extract was transferred to a 2 mL Eppendorf tube, and then 400 μL was filtered through 0.2 μm at 6000 g for 1 min at 4 °C before direct analysis by LC-MS/MS [45,62]. For every 1 mL of methanolic extract, 125 μL of 2.5 M NaOH was added. The whole mixture was put in a vortex for 30 s and then heated for 40 min at 76 °C. After cooling in an ice bath, the mixture was neutralized and mixed with 125 μL of 2.5 M HCl. All samples were filtered (0.2 μm, Nanosep, MF, Pall) and stored at 20 °C before analysis.

##### LC-MS/MS Conditions

The chosen analytical method was obtained after a phase of optimization of the different liquid chromatography (LC) parameters and MS/MS detection. LC-MS/MS analysis was performed on an LC system (UFLC XR, Shimadzu, Marne La Vallee, France) coupled to a hybrid triple quadrupole/linear ion-trap mass spectrometer (API 4000 Qtrap, Sciex, Villebon sur Yvette, France) equipped with a heated electrospray ionization (ESI) source.

Lipophilic toxins were separated on a Kinetex XB-C_18_ (100 × 2.1 mm, 2.6 µm) (Phenomenex) with its pre-column maintained at 40 °C with a 0.3 mL/min flow rate and an injection volume of 5 µL. Mobile phases consisted of water (A) and methanol/water (95:5, *v*/*v*) (B) both containing 2 mM ammonium formate and 50 mM formic acid. This approach required the use of three LC-MS/MS methods (i.e., three elution gradients and two ionization modes) [45,62] (Appendix A). MS/MS analysis of all toxins was carried out with multiple reaction monitoring detections (MRM) in three sequences: two in positive mode and one in negative mode with optimized source parameters (Appendix A). The optimized MRM parameters (declustering potential, collision energy, and collision cell exit) are reported in Appendix A.

Ionization recoveries: the matrix effects on electrospray ionization of toxins were estimated by spiking the extract of uncontaminated mussels with the toxins standards. Quantification was performed relative to the toxins standard using a six-point calibration curve. The LC-MS/MS results of the field samples analyzed are corrected by ionization and extraction recovery.

#### 4.3.2. HILIC-MS/MS Analysis of STXs, TTXs, ATXs, and CYN Toxins Groups

The method is based on Turner et al. (JAOAC 2020) [91], with slight modifications and optimized to be applicable to hydrophilic cyanotoxins. It has been fully validated in-house.

##### Sample Preparation

5 ± 0.1 g of homogenized shellfish tissue weighed into a 50 mL polypropylene centrifuge tube were extracted with 5 mL of 1% acetic acid by shaking on a vortex mixer for 90 s. The tubes were then placed in a boiling water bath for 5 min, cooled to room temperature in a cold water bath, vortexed again, and centrifuged at 9000× *g* for 10 min. The supernatant was transferred to a 50 mL graduated polypropylene tube while the pellet was extracted a second time with 10 mL of 1% acetic acid and centrifuged at 9000× *g* for 10 min. The supernatant was then decanted into the same graduated polypropylene tube that contained the first portion of the extract, and the combined extracts were adjusted to 20 mL with 1% acetic acid.

4 mL of the supernatants adjusted to 20 mL were transferred to a 15 mL polypropylene tube, and 20 µL of 25% NH_3_ (LC-MS grade) was added. After vortexing, 1 mL was transferred to a 1.5 mL polypropylene centrifuge tube and centrifuged at 14,500× *g* for 1 min. 800 µL of centrifuged extract was cleaned up manually through 250 mg/3 mL solid phase extraction cartridges of amorphous polymer graphitized carbon. The cartridges were conditioned with 3 mL of 30% acetonitrile + 1% acetic acid, followed by 3 mL of 0.025% NH_3_, and 800 µL of extract was added to the conditioned cartridges. Then, it was washed with 700 µL of ultra-pure water and discarded to waste. PSTs, TTXs, and hydrophilic cyanotoxins were then eluted with 2 mL of 30% acetonitrile + 1% acetic acid in a 5 mL polypropylene tube. The eluate was diluted 1 to 4 with acetonitrile and filtered through nylon filters of 0.20 µm porosity before direct analysis by LC-MS/MS [92].

##### HILIC-MS/MS Conditions

The LC system was an Accela 1250 (Thermo Fisher Scientific, San Jose, CA, USA). Ultra-high-performance liquid chromatographic (UHPLC) separation of toxins was performed using an Acquity UPLC BEH Amide column or an Acquity UPLC BEH Glycan column (Waters, Milford, MA, USA). Both columns had similar dimensions (150 × 2.1 mm, 1.7 μm particle sizes, 130 Å) and each was equipped with its specific Vanguard pre-column system (5 × 2.1 mm, 1.7 µm particle sizes, 130 Å). The column temperature was set to 70 °C. Two injections with different separation conditions were applied to detect all the toxins sought. For PSTs and TTXs analysis (LC conditions A), eluent A was composed of water/formic acid/ammonia 250:0.0375:0.15 (*v*/*v*/*v*), and eluent B of acetonitrile/water/formic acid 700:300:0.1 (*v*/*v*/*v*). For ATXs and CYNs (LC conditions B), eluent A was also composed of water/formic acid/ammonia 250:0.0375:0.15 (*v*/*v*/*v*), and eluent B of acetonitrile/water/formic acid 900:100:0.1 (*v*/*v*/*v*). The elution gradients for LC conditions A and B are shown in Appendix A. For both conditions, A and B, 2 µL were injected into the system.

The detection of the toxins was performed with a TSQ Vantage triple quadrupole mass spectrometer (Thermo Fisher Scientific), equipped with an electrospray ionization (ESI) source (HESI-II probe). The mass spectrometer was operated in selected reaction monitoring (SRM) mode in positive and negative ionization modes. For both methods, the source parameters were the same. The spray voltage was 3000 V in negative ionization mode and 4000 V in positive ionization mode. The source temperature was set at 500 °C and the capillary temperature at 350 °C. Nitrogen was used as the nebulizing gas with a sheath gas pressure of 60 (arbitrary units) and an auxiliary gas pressure of 20 (arbitrary units). The collision gas was argon, with a gas pressure of 1.5 mTorr. One transition was used for quantification (Q) and another as a qualifier transition (q). The optimized compound-dependent parameters are listed in Appendix A. A mass resolution of 0.7 Da (full width at half maximum) was set for the first and third quadrupoles (Q1 and Q3). Matrix-matched calibration curves were used as external standards for quantification, as described by Turner et al. (JAOAC 2020) [91]. They were prepared by dilution of a mixed stock solution in a toxin-free shellfish tissue after extraction by 1% acetic acid, solid phase extraction through 250 mg/3 mL cartridges of amorphous polymer graphitized carbon, and dilution with acetonitrile.

#### 4.3.3. HILIC-MS/MS Analysis of BMAA and Its Analogues (DAB, AEG)

##### Sample Preparation

BMAA, DAB, and AEG (total form) were extracted and analyzed as previously described [54]. Briefly, 750 µL of 0.1 M TCA containing the isotopically labeled internal standards D_3_BMAA and D_5_DAB (77.5 ng mL^−1^) were added to 100 mg of a total shellfish meat grind before grinding with glass beads (Ø 150–250 µm) in a mixer mill (Retsch MM400, Hamburg, Germany) for 2 min. After centrifugation at 15,000× *g* for 5 min, 150 µL of the supernatant was subsequently collected and evaporated to dryness. The residue was dissolved in 600 µL HCl 6 M and hydrolyzed at 99 °C for 21 h. HCl was dried at 50 °C under a flux of nitrogen, and the residue was dissolved in 1 mL of 0.1 M TCA before SPE clean-up. The cation-exchange cartridge (Plexa PCX, Agilent, 60 mg) was previously conditioned with 2 mL methanol (MeOH) and 1 mL TCA 0.1 M. After loading the sample, the cartridge was rinsed with 1 mL hydrochloric acid (HCl) 0.1 M and 2 mL MeOH before elution with 4 mL of a mixture of MeOH and ammonium hydroxide at 25% NH_3_ (93:7, *v*/*v*). The eluate was evaporated to dryness at 50 °C under a flux of nitrogen, and the residue was solubilized in a mixture of acetonitrile/water (63:37, *v*/*v*), both containing 0.1% formic acid (FA), before analysis by HILIC-MS/MS.

##### HILIC-MS/MS Conditions

Analysis was performed by liquid chromatography coupled to tandem mass spectrometry (LC-MS/MS) on an Ultra-Fast Liquid Chromatography system (UFLC) (Nexera, Shimadzu) coupled to a triple-quadrupole mass spectrometer (5500 QTRAP, Sciex) [54]. Chromatography was performed with a ZIC^®^-HILIC column (100 × 2.1 mm, 3.5 μm, Merck Sequant^®^, Darmstadt, Germany) with a suited guard column. Mobile phases were Milli-Q water (Mobile Phase A) and acetonitrile (Mobile Phase B), both containing 0.1% formic acid. The flow rate was 0.25 mL/min, and the injection volume was 5 μL. The column temperature was 60 °C, while samples were kept at 4 °C. The linear gradient elution started with 37% of Mobile Phase A, rising to 60% over 10 min, held for 0.5 min, then decreased to 37% of Mobile Phase A and held for 5 min to equilibrate the system. The HILIC-MS/MS system was used in positive ion mode with multiple reaction monitoring (MRM) detections (Appendix A). The MS/MS detection was performed in positive ionization mode using multiple reaction monitoring (MRM). The source parameters were curtain gas 20 psi, temperature 600 °C; Gas-1: 40 psi, Gas-2: 60 psi, and ion spray voltage 5500 V. The optimized MRM parameters (declustering potential, collision energy, and collision cell exit) are reported in Appendix A.

BMAA and DAB were unambiguously distinguished thanks to chromatographic resolution, specific mass spectral transitions, and qualitative-to-quantitative ion ratios. BMAA and DAB were quantified using external calibration curves of pure standards constructed by dilution series of stock solutions in a mixture of can and water (63/37, *v*/*v*, both containing 0.1% FA). A corrective factor derived from D_3_BMAA and D_5_DAB recoveries was applied for more accurate quantification.

## 5. Conclusions

All the results obtained in this study show the presence of phycotoxins not regulated in Europe and of freshwater cyanotoxins in French marine shellfish. These toxins include spirolides (SPX-13-DesMeC, SPX-DesMeD), pinnatoxins (PnTX-G, PnTX-A), and BMAA/DAB, already detected since 2006, 2010, and 2014, respectively, before EMERGTOX was set up. Other toxins were identified for the first time by the EMERGTOX network: gymnodimines (GYM-A), brevetoxins (BTX-2, BTX-3), microcystins (dmMC-RR, MC-RR), CYN, and ATX. Although some toxins were not detected (e.g., TTX), their research will continue over the long term to monitor their emergence at the national level.

Our results revealed the presence of emerging toxins in bivalve shellfish collected on French coasts, raising questions about the future needs for French safety regulatory monitoring programs, in particular for (i) brevetoxins, which represent a proven human health risk and are regulated outside Europe (USA, Australia, New Zealand, Mexico); and (ii) MCs, for which there is a need to establish a guidance level in shellfish in order to protect human health. However, the bioavailability of bound MCs in shellfish tissues is still an issue and deserves further research.

To facilitate data access and interpretation, a web interface is currently being developed. This new tool is expected to allow the identification of contamination trends and facilitate the interpretation of the data, thus allowing a fine analysis of the health situation and its evolution. For example, the tool will provide graphical visualizations of the results showing temporal evolution at different time scales or sampling sites per toxin analog and the probability of exceeding the guidance level for unregulated toxins in French shellfish.

## Figures and Tables

**Figure 1 marinedrugs-21-00435-f001:**
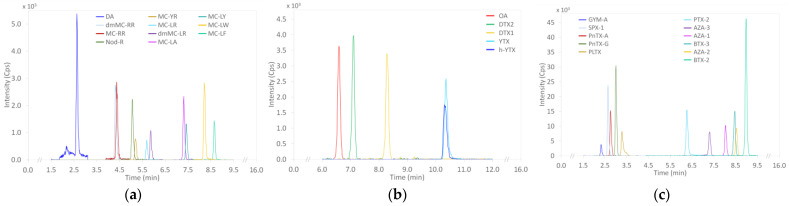
Examples of chromatograms obtained for each of the three analytical methods after doping of the mussel matrix with (**a**) DA, MCs, NOD; (**b**) OA/DTXs, YTXs; (**c**) GYM-A, SPX-13-desMe-C, PnTXs, PLTX, PTX-2, AZAs, BTXs.

**Figure 2 marinedrugs-21-00435-f002:**
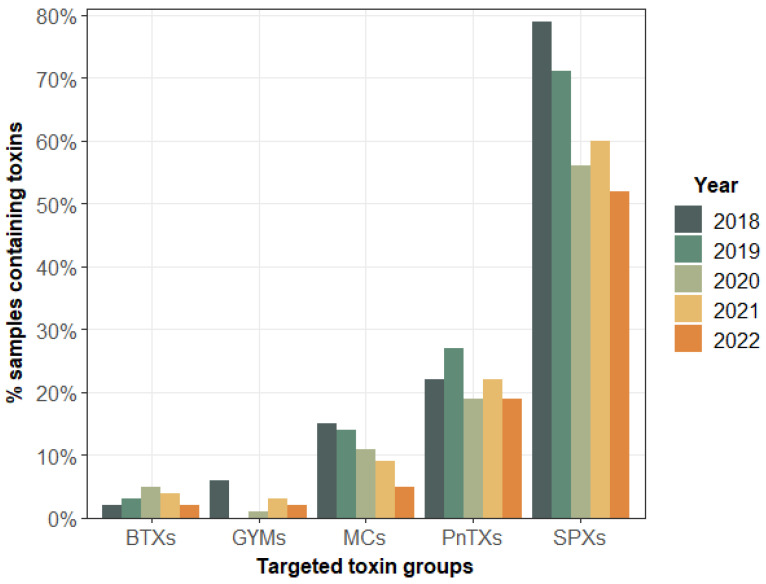
Percentage of shellfish samples from the EMERGTOX network containing different groups of unregulated lipophilic phycotoxins and microcystins quantified during the period 2018–2022.

**Figure 3 marinedrugs-21-00435-f003:**
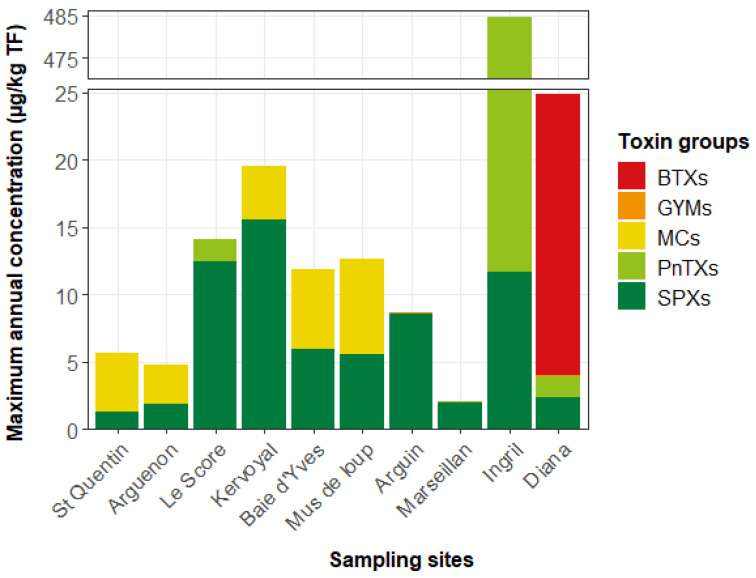
Maximum concentrations of unregulated toxins found in shellfish at different sites of the EMERGTOX network in 2018.

**Figure 4 marinedrugs-21-00435-f004:**
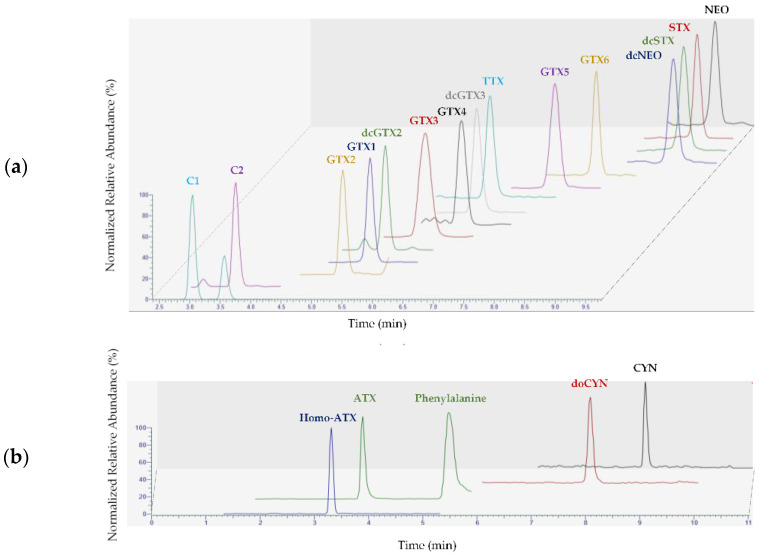
Chromatograms obtained with (**a**) HILIC conditions A for a blank mussel matrix spiked with PSPs and TTX; (**b**) HILIC conditions B for a blank mussel matrix spiked with ATXs and CYNs.

**Figure 5 marinedrugs-21-00435-f005:**
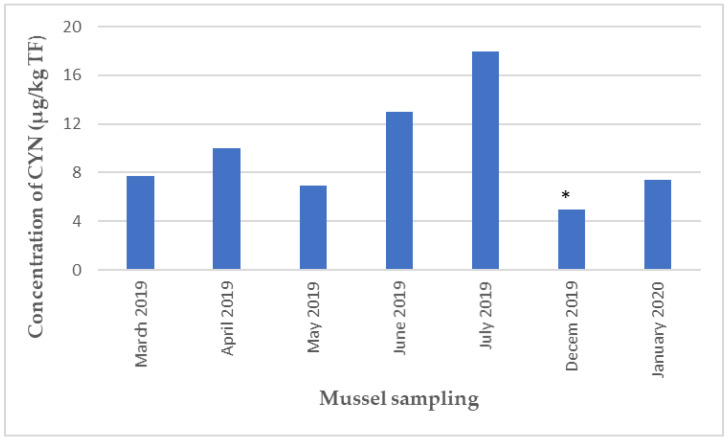
Concentration of CYN in mussels collected at the Ingril site on the Mediterranean coast during the period 2019–2020. * In order to show the presence of toxins at levels between the LOD and LOQ on the graph, the LOQ value of CYN was attributed to these specific cases.

**Figure 6 marinedrugs-21-00435-f006:**
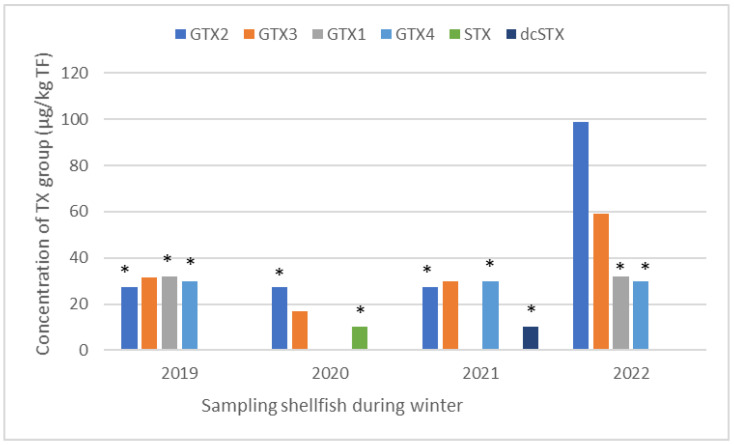
Maximum concentration of the saxitoxin group in oysters collected at the Arguin site (Atlantic coast) during winters in the period 2019–2022. * In order to show the presence of toxins at levels between the LOD and LOQ on the graph, the LOQ value for each toxin detected was attributed to these specific cases.

**Figure 7 marinedrugs-21-00435-f007:**
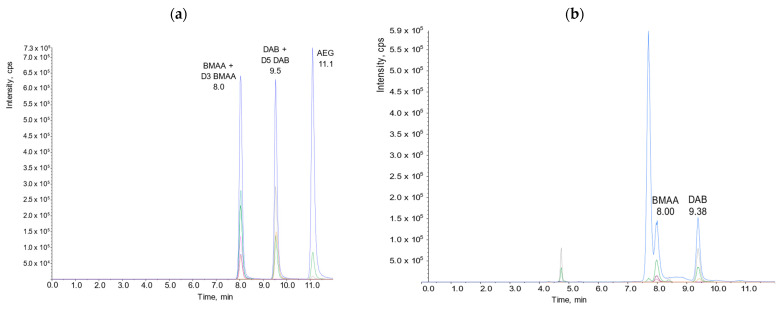
HILIC-MS/MS analysis of (**a**) a mixture of BMAA, D3-BMAA, DAB, D5-DAB, and AEG standards; (**b**) Oyster sample collected at Baie des Veys.

**Figure 8 marinedrugs-21-00435-f008:**
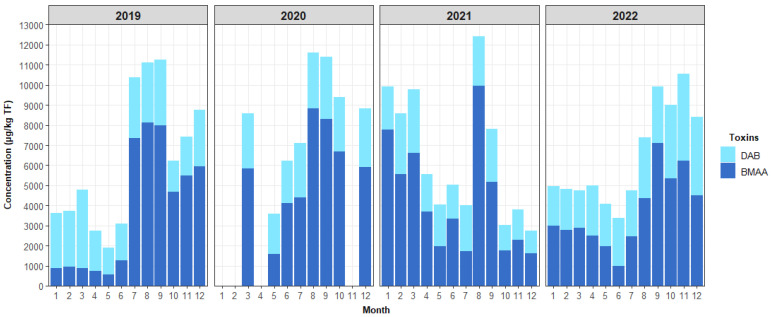
Evolution of the monthly concentration of BMAA and DAB in mussels at the Kervoyal site (Atlantic coast) during the 2019–2022 period.

**Figure 9 marinedrugs-21-00435-f009:**
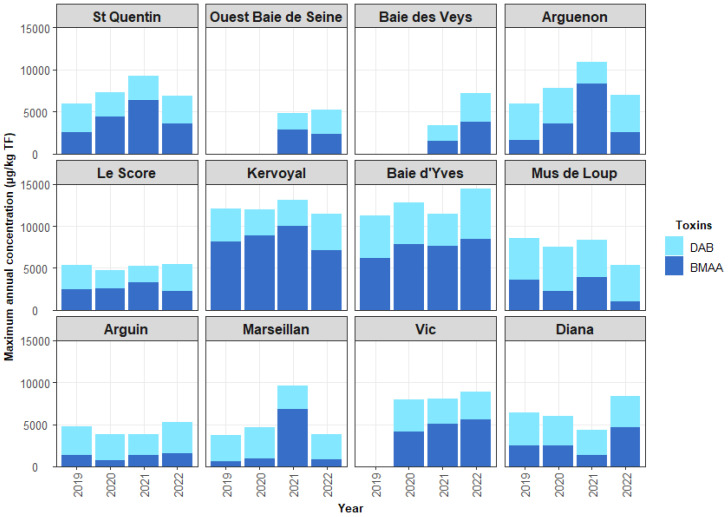
Maximum concentrations of BMAA and DAB per year and per site during the 2019–2022 period.

**Figure 10 marinedrugs-21-00435-f010:**
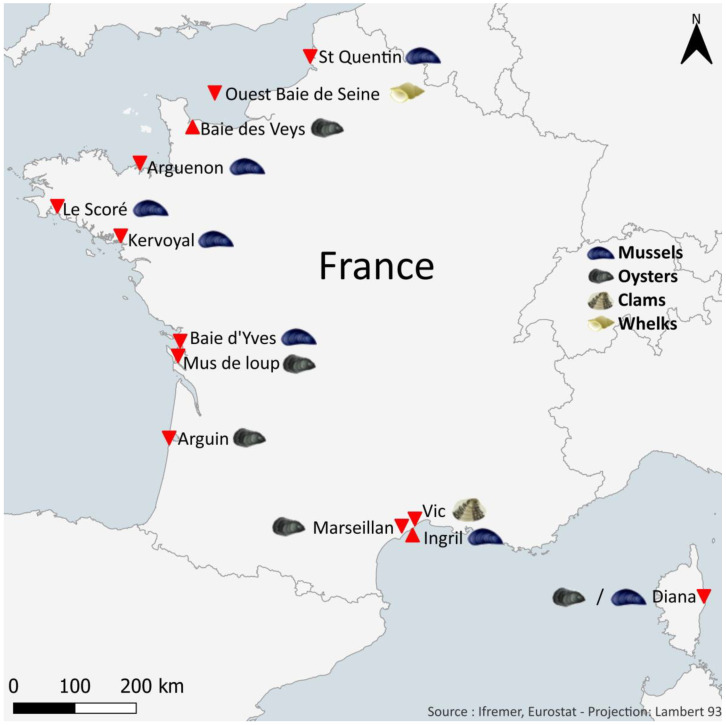
Map of the shellfish sampling sites in France used within the EMERGTOX network. Sites are indicated by red triangles; shellfish species studied at these localities are indicated by symbols defined in the legend.

## Data Availability

The data presented in this study are available on request from the corresponding author. The data will be publicly accessible in the Ifremer Quadrige database as soon as they are referenced.

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
