# Peer review of "Five Years Monitoring the Emergence of Unregulated Toxins in Shellfish in France (EMERGTOX 2018–2022)"

_marinedrugs, 2023, doi:10.3390/md21080435_

Round 1

Reviewer 1 Report

It is great to see the results from 5 years of monitoring this wide range of emerging toxins in a single paper. This information is valuable for any risk assessment for these toxins in French shellfish. It has also identified areas where further work is required to improve analytical methods, particularly for the palytoxins and brevetoxins groups.

I have the following minor comments and suggestions:

1.       L36 - The first sentence in the introduction describes the three grougs of marine toxins that are monitored. It is not clear from the way they are list which are the three distinct groups, as it starts with ‘lipophilic toxins’ and then lists OA, DTXs, YTXs, AZAs, STX, DA. Maybe just state – lipophilic toxins, saxitoxins and domoic acid.

2.       L59 – I think it is worth mentioning here that some are quite toxic orally to mice e.g. pinnatoxins.

3.       L63 – change hydroxyl to hydroxyls

4.       L65 – Is Palythoa a coral? I’m not 100% sure of the definition of a coral, please check.

5.       L73 – TTX doesn’t ‘block a receptor’, it binds to a receptor at site 1 on voltage gated sodium channels which blocks the channel.

6.       Figure 2 – The abbreviations for domoic acid (DA) and okadaic acid (OA) and have had letters reversed to AD and AO, both in the figure and the description

7.       L306 – What is meant by the LOD equalled the LOQ? Also equalled was spelt incorrectly (should have two l’s).

8.       There is no mention of BAMA, an isomer of BMAA. This is a potential interference, have you confirmed that BAMA is separated from BMAA on this method? If so, is there evidence that can be provided? I noticed there is a large unidentified peak in the chromatogram shown in figure 8, is this BAMA?

9.       L522 – change chlorohydric acid to hydrochloric acid or maybe just HCl to be consistent with the rest of the text.

10.   L557 – I think the term pellet should be used to replace sediment

11.   L562 - ..to a 15mL polypropylene tube of 15 mL…

12.   L606 – change durant to for

13.   L607 - ..collected, and evaporated

14.   L613 – what was the concentration of ammonium hydroxide used?

15.   L643 – The sentence about other toxins being identified for the first time is not correct. BMAA had been previously reported (see ref 54).

16.   Table S1 – change LD to LOD and LQ to LOQ

17.   Table S2 – Change 17,5 to 17.5

18.   Table S6 – The use of 2 d.p. for the concentrations of the standards indicates more precision than is possible e.g. 6470.00 ng/mL. I would suggest changing to significant figures maybe 2 or 3 sig figs.

19.   Table S8 – change commas to decimals for anatoxin ions e.g. 166,1 to 166.1

20.   Figure 2 and Table S5 – The abbreviation SPX1 is not described anywhere, I recommend changing to SPX-13-desMe-C

21.   There is a bit of editing required mostly in the methods and materials.

·         Use subscript for number of deuterium’s on internal standards e.g. D3BMAA or BMAA-d3

·          Formatting of units °C, mL not ml etc

·         Consistency of reagent names/abbreviations NH3 vs ammonium hydroxide, hydrochloric vs HCl

·         I would suggest stating the concentration of a solution prior to the reagent name e.g. 0.1 M TCA rather than TCA 0.1 M, this would be more consistent with other examples in the document 2.5 M NaOH or 1% acetic acid.

The English language used was very understandable. There are a few spelling mistakes, and some minor grammatical improvements could be made. More concise sentences could be used in places, but overall it is very readable.

Reviewer 2 Report

1.        It is suggested that some results of the determination of toxins in shellfish samples be added to the Abstract.

2.        The keywords are too many and it is suggested to delete some of them.

3.        Line 47, BMAA is also widespread in the marine environment and can be produced by marine diatoms, so it is not recommended that BMAA be directly classified as a freshwater cyanotoxin.

4.        Part 2.1 is proposed to be placed in the methods section;

5.        In the Results section, parameters such as toxin recoveries, matrix calibration curves etc. should be given in the manuscript.

6.        In the results section, the concentrations of toxins in each shellfish should be presented in Tables.

7.        Line 223, No Figure S2 is present in the Annex material.

8.        In the discussion section, the current contamination of French shellfish with phycotoxins should be discussed in comparison with that of other EU countries.

9.        The units of temperature are misspelled in several places in the text.

Round 2

Reviewer 2 Report

The manuscript has been revised in accordance with my suggestions and is recommended for acceptance for publication.